Sagittal spinal morphotype assessment in 8 to 15 years old Inline Hockey players

Sainz de Baranda Pilar 1
Cejudo Antonio 1 antonio.cejudo@um.es
Moreno-Alcaraz Victor Jesus 1
Martinez-Romero Maria Teresa 1
http://orcid.org/0000-0002-1655-4169 Aparicio-Sarmiento Alba 1
http://orcid.org/0000-0002-3593-3287 Santonja-Medina Fernando 2
1 Department of Physical Activity and Sport/Faculty of Sport Sciences/Campus of Excellence Mare Nostrum, Universidad de Murcia , Murcia , Spain
2 Department of Surgery, Pediatrics, Obstetrics and Gynecology, “Virgen de la Arrixaca” University Hospital/Faculty of Medicine/Campus of Excellence Mare Nostrum, Universidad de Murcia , Murcia , Spain
Gao Liang
Electronic publication date: 2020 Jan 2
Publication date: 2020
Volume: 8
Electronic Location ID: e8229
Received 2019 Jul 19; Accepted 2019 Nov 18
Copyright: © 2020 Sainz de Baranda et al.
Copyright year: 2020
Copyright holder: Sainz de Baranda et al.
License: This is an open access article distributed under the terms of the Creative Commons Attribution License, which permits unrestricted use, distribution, reproduction and adaptation in any medium and for any purpose provided that it is properly attributed. For attribution, the original author(s), title, publication source (PeerJ) and either DOI or URL of the article must be cited.
License URL: https://creativecommons.org/licenses/by/4.0/

Keywords: Spine, Alignment, Morphotype, Young athlete, Injury

Funding: Spanish Ministry of Economy, Industry and Competitiveness DEP2010-21793 Spanish Ministry of Education, Culture and Sports for the Training of University Teaching Staff FPU15/05200 and FPU18/00702 Study of injury risk in young athletes through artificial intelligence networks Spanish Ministry of Science and Innovation DEP2017-88775-P Facoltà di Scienze Motorie of Università degli Studi di Urbino “Carlo Bo” from 29/10/2018 to 03/02/2019 The research was supported by the Spanish Ministry of Economy, Industry and Competitiveness (Grant Number: DEP2010-21793); and the Spanish Ministry of Education, Culture and Sports for the Training of University Teaching Staff (Grant Numbers: FPU15/05200 and FPU18/00702). This research is part of the project entitled “Study of injury risk in young athletes through artificial intelligence networks,” funded by the Spanish Ministry of Science and Innovation (reference: DEP2017-88775-P). This study was carried out during the research stay at the Facoltà di Scienze Motorie of Università degli Studi di Urbino “Carlo Bo” from 29/10/2018 to 03/02/2019 funded by Erasmus Teaching. The funders had no role in study design, data collection and analysis, decision to publish, or preparation of the manuscript.

==============================
Background

Physiological sagittal spinal curvatures play an important role in health and performance in sports. For that reason, several scientific studies have assessed spinal morphology in young athletes. However, to our knowledge, no study has assessed the implications of Inline Hockey (IH) practice on sagittal integrative spinal morphotype in adolescent players.

Objectives

The aims of the present study were to describe habitual sagittal spinal posture in young federated IH players and its relationship with training load and to determine the sagittal integrative spinal morphotype in these players.

Methods

An observational analysis was developed to describe the sagittal spinal morphotype in young federated IH players. A total of 74 IH players from the Technification Plan organized by the Skating Federation of the Valencian Community (aged from 8 to 15 years) participated in the study. Thoracic and lumbar curvatures of the spine were measured in a relaxed standing position (SP), in a slump sitting position (SSP) and in maximum flexion of the trunk (MFT) to determine the “Sagittal Integrative Morphotype” of all players. An unilevel inclinometer was used to quantify the sagittal spinal curvatures. The Hip Joint Angle test was used to quantify the Lumbo-Horizontal angle in flexion (L-H fx) of all participants with a goniometer.

Results

When thoracic curvature was analyzed according to normality references, it was found that 64.9% of IH players had thoracic hyperkyphosis in a SSP, while 60.8% and 74.3% of players were classified as normal in a SP and in MFT, respectively. As for the lumbar curve, 89.2% in a SP and 55.4% in MFT were normal, whereas 68.9% of IH players presented lumbar hyperkyphosis in a SSP. Regarding the “Sagittal Integrative Morphotype,” only 17.6% of players were classified as “Normal” in the three measured positions for the thoracic curve, while 37.8% had “Thoracic Hyperkyphosis” and 41.8% presented “Functional Thoracic Hyperkyphosis.” As for the “Sagittal Integrative Lumbar Morphotype,” only 23% of athletes had a normal curve in the three positions, whereas 66.2% presented “Functional Lumbar Hyperkyphosis.” When the L-H fx was evaluated, the results showed that only 16.2% of the athletes were classified as normal.

Conclusions

Federative IH practice seems to cause specific adaptations in spinal sagittal morphotype. Taking into account the “Sagittal Integrative Morphotype” only 17.6% IH players presented “Normal Morphotype” with a normal thoracic kyphosis in the three measured positions, while only 23% IH players presented “Normal Morphotype” with a normal lumbar curvature in the three assessed positions. Furthermore, only 16.2% of IH players showed normal pelvic tilt. Exercise programs to prevent or rehabilitate these imbalances in young IH players are needed.

Introduction

Physiological sagittal spinal curvatures play an important role in health and performance in sports since the distribution of mechanical strains greatly affects the structures of the spine and can influence athletes’ stability as well as result in overuse injuries to the spine (Keller et al., 2005). Hence, these curvatures should be neither reduced nor excessive in order to maintain a physiological, harmonic and balanced posture. In this sense, to have sagittal spinal curvatures within the normal ranges could favor the athlete’s trunk mobility as well as improve a player’ stability due to the lower center of gravity and the better distribution of the load (Ackland, Elliott & Bloomfield, 2009).

It must be noted that sagittal misalignments of the spine alter the loads distribution and increase even more the stress on the different joint tissues; therefore, an unbalanced sagittal spine predisposes to back problems. Previous studies have found that an increased thoracic or lumbar curvature has been related to spinal pain (Christie, Kumar & Warren, 1995; Ohlén, Wredmark & Spangfort, 1989; Roncarati & McMullen, 1988; Salminen et al., 1992; Salminen et al., 1993), as well as to certain pathologies in the spine (Katz & Scerpella, 2003; Swärd et al., 1990). For instance, it has been observed instability of the spine in kyphotic lumbar postures (Green, Grenier & McGill, 2002; Jackson et al., 2001; Solomonow et al., 1999), disc protrusion in hyperkyphotic postures (Callaghan & McGill, 2001; Simunic, Broom & Robertson, 2001), herniated disc when the lumbar curve is inverted or kyphotic (Micheli & Trepman, 1990), anterior vertebral wedges (Santonja & Martínez, 1995), Schmorl nodules or vertebral plate abnormalities (Callaghan & McGill, 2001; McGill, 2002) in hyperkyphotic and inverted positions of the lumbar spine, and facet degeneration and spondylolysis in hyperlordotic postures (Micheli & Trepman, 1990). These negative consequences justify the research on the relationship between systematic sports training and the alignment of sagittal spinal curvatures (Santonja & Morales, 2008).

For those reasons, several experts in the analysis of the locomotor system recommend the assessment of sagittal spinal curvatures to describe the sagittal morphotype of spine in sports (Sainz de Baranda, Rodríguez-García & Santonja, 2010; Sainz de Baranda, Santonja & Rodríguez-Iniesta, 2009; Santonja & Pastor, 2000; Sanz-Mengibar, Sainz-de-Baranda & Santonja-Medina, 2018). In fact, this knowledge could contribute to the development of more effective preventive interventions to be adopted by a multidisciplinary professional team. Specifically, for the assessment of the sagittal plane of the spine, it is recommended to evaluate the thoracic and lumbar curves in a SP, in a SSP and in MFT to finally establish the “Sagittal Integrative Morphotype” of the spine (Santonja, 1996).

In addition, as the spine of an adolescent is in a maturation period, it shows changes in posture and balance of its curvatures during growth and it is more vulnerable (Sainz de Baranda et al., 2006). Thus, sports professionals should be aware of the loads and overloads inherent in sport and training and its impact on the young athlete’s spine (Sainz de Baranda, Rodríguez-García & Santonja, 2010).

Therefore, several scientific studies have assessed spinal morphology in young athletes as professional soccer players (Sainz de Baranda et al., 2001), basketball players (Ferreira-Guedes & Amado-João, 2014; Grabara, 2016a), handball players (Grabara, 2014a), volleyball players (Grabara, 2015), rhythmic gymnasts (Martínez-Gallego & Rodríguez-García, 2005; Ohlén, Wredmark & Spangfort, 1989), swimmers (Pastor et al., 2002; Santonja & Pastor, 2000), dancers of Spanish and Classical dance (Gómez-Lozano, 2007), cricket players (Hecimovich & Stomski, 2016), cross-country skiers (Alricsson et al., 2016) and wrestling (Rajabi et al., 2008). Other studies, included athletes of different sports (Betsch et al., 2015; Grabara, 2014a; Lichota, Plandowska & Mil, 2011; Wojtys et al., 2000). However, to the best of our knowledge, no study has assessed the implications of Inline Hockey (IH) practice on “Sagittal Integrative Spinal Morphotype” in adolescent players.

The participation in Inline roller hockey or IH among adolescents has increased in the past few years thanks to the popularity of inline-skating. Since its introduction in 2000 in Spain, IH has been one of the fastest growing sports in the different federative categories. Over the 2005/06 and 2016/17 seasons, there were 3,100 and 5,234 licenses, respectively, which is an increase of almost 60% in the number of licenses within the last 10 years (Real Federación Española de Patinaje, 2018).

In order to find out how IH can affect young players’ spine as well as to help sport professionals to plan specific preventive interventions, the current investigation was carried out. The aims of the present study were: (1) to describe habitual sagittal spinal posture in young federated IH players and its relationship to training load, and (2) to determine the “Sagittal Integrative Spinal Morphotype” in these players. Our hypothesis is that there is a special adaptation of the spine to the specific requirements of IH in young players.

Materials and Methods

In order to confirm or rule out our hypothesis, an observational analysis was developed to describe the sagittal spinal morphotype in young federated IH players.

The study was approved by the Ethics and Research Committee of the University of Murcia (Spain) (ID: 1702/2017).

Participants

The subject population was selected through a convenience sample from the Technification Plan organized by the Skating Federation of the Valencian Community in the season 2016–2017, in which the best IH players of the Valencian Community took part in (Castellón de la Plana, Region of Valencia, Spain). A total of 90 IH players from the Skating Federation of the Valencian Community were selected to participate in this study.

Following the inclusion criteria, those who were from 8 to 15 years old and were playing within the Spanish Federative Categories of “Benjamín” (U11) “Alevín” (U13) and “Infantil” (U16) were included in the study (n = 77), whereas goalkeepers and players who belonged to the U17 team were not included (n = 15). However, those who had previously received treatment for any frontal or sagittal plane-related pathology by the use of a corset or specific kinesiotherapy or those who presented specific symptoms or musculoskeletal limitations to perform the tests correctly were excluded (n = 1).

Finally, a total of 74 IH players U16 participated in the study (Table 1).

Table 1 Demographic and training data of the U16 IH players (n = 74)*.

	Minimum	Maximum	Mean ± SD	
Age (years)	8.0	15.0	12.1 ± 1.8	
Body weight (kg)	27.0	86.1	51.5 ± 12.7	
Height (cm)	1.30	1.83	1.55 ± 0.12	
BMI (kg/m2)	15.0	28.6	21.1 ± 3.4	
Training hours per year	72.0	308.0	164.45 ± 49.95	
Training volume	96.0	2,160.0	608.67 ± 469.46	
Stick length (cm)	108.0	155.0	134.1 ± 10.4	
Note:

* SD, standard deviation; BMI, body mass index.

Procedure

The study was conducted in the season 2016–2017. According to the Declaration of Helsinki, the procedures and potential risks were explained to IH players, parents and coaches prior to participation and legal tutors expressed written consent.

“Sagittal Integrative Morphotype,” as well as Lumbo-Horizontal angle in flexion (L-H fx) of all participants, were assessed. In addition, participants completed an ad hoc questionnaire about their sport-related background (federative category, current competitive level, tactical position, stick length, dominant leg (defined as the participant’s preferred kicking leg)), anthropometric characteristics (age, weight, height and body mass index), regular training workload (years of sport experience, training months per year, training days per week, training hours per week, current competitive level) as well as about prior and current musculoskeletal injuries and treatment.

According to Wojtys et al. (2000) and Sainz de Baranda, Rodríguez-García & Santonja (2010), “training hours per year” and “training volume” were calculated. “Training hours per year” was equal to training hours per week × 4 weeks per month × months per year. “Training volume” was equal to years of sport experience × training hours per year. Two groups were established based on those who had trained more or less than 160 h per year. The players were also divided according to the accumulated training workload in more or less than 480 h of training.

Sagittal spinal morphotype assessment

Data from each IH player were taken during the same assessment session and with the same temperature (25 °C). All the measurements were performed by the same Sport Science expert and participants were assessed wearing undergarments and barefoot. Athletes did not perform warm-up or stretching exercises before or during the measurement in order to achieve real clinical conditions (Aalto et al., 2005; Cejudo, 2015; Ginés-Díaz et al., 2019).

An unilevel inclinometer (ISOMED, Inc., Portland, OR, USA) was used to quantify the sagittal spinal curvatures by providing considerable reproducibility and validity, with a good correlation with the radiographic measurement (Mayer et al., 1984; Saur et al., 1996), and according to the methodology described by Santonja (1996), which has been used in previous studies (Ginés-Díaz et al., 2019; Sainz de Baranda, Santonja & Rodríguez-Iniesta, 2009; Sanz-Mengibar, Sainz-de-Baranda & Santonja-Medina, 2018). A goniometer provided with a spirit level system was used to quantify the L-H fx (Sainz de Baranda et al., 2014; Santonja, Andújar & González-Moro, 1994).

Intra-tester reliability of thoracic and lumbar curvatures and pelvic tilt was calculated in a previous pilot study. A priori reliability was established by the primary investigator in a sample of convenience (university students; n = 12, ranged from age = 20–22 years; (age: 25.8 + 0.9 years; height: 1.71 + 0.09 m; body mass: 72.05 ± 9.28 kg) measured on two occasions by the same tester in a single session. Intra-class correlation coefficients (ICC) with 95% confidence intervals (CI) were calculated. The ICC and the minimal detectable change at a 95% confidence interval (MDC95) values for all measures ranged from 0.88 to 0.97 (95% CI [0.70–0.99]) and 0.53° to 1.1° respectively (95% CI [0.38°–1.6°]).

The assessment protocol of “Sagittal Integrative Morphotype” as defined by Santonja (1996) is composed by the evaluation of sagittal spinal curvatures in a relaxed SP (Fig. 1A), in a SSP (Fig. 1B) (Sainz de Baranda et al., 2006; Sainz de Baranda, Santonja & Rodríguez-Iniesta, 2009; Sainz de Baranda, Santonja & Rodriguez-Iniesta, 2010; Santonja, 1996) as well as in MFT (Fig. 1C) (López-Miñarro et al., 2007; Sainz de Baranda, Rodríguez-García & Santonja, 2010; Sainz de Baranda, Santonja & Rodríguez-Iniesta, 2009; Sanz-Mengibar, Sainz-de-Baranda & Santonja-Medina, 2018). This protocol is performed in order to have a more accurate diagnostic of sagittal spinal morphotype (López-Miñarro et al., 2007; Norkin & White, 1995; Sainz de Baranda, Rodríguez-García & Santonja, 2010; Sainz de Baranda, Santonja & Rodríguez-Iniesta, 2009; Sanz-Mengibar, Sainz-de-Baranda & Santonja-Medina, 2018).

Figure 1 Assessment positions for the “Sagittal Integrative Morphotype” protocol.

(A) SP. (B) SSP. (C) MFT.

Prior to data collection, the spinous process of the first thoracic vertebra (T1), 12th thoracic vertebra (T12) and 5th lumbar vertebra (L5–S1) were marked on the skin of participants (López-Miñarro et al., 2007; Norkin & White, 1995; Sainz de Baranda, Rodríguez-García & Santonja, 2010; Sainz de Baranda, Santonja & Rodríguez-Iniesta, 2009; Sanz-Mengibar, Sainz-de-Baranda & Santonja-Medina, 2018).

Standing position

To assess the SP, the participant was standing and relaxed (Ginés-Díaz et al., 2019; Sanz-Mengibar, Sainz-de-Baranda & Santonja-Medina, 2018). The inclinometer was placed at the first mark (T1) and calibrated to 0°, then the curvature was profiled until maximum angulation of thoracic curvature was reached and the angle was recorded. Subsequently, the inclinometer was calibrated to 0° again at this point and the lumbar curvature was profiled until the maximum angle was reached and recorded.

Slump sitting position

To measure the SSP, the participant was sitting on the stretcher in a relaxed posture with the forearms resting on the thighs, knees flexed and without feet support (Ginés-Díaz et al., 2019; Sanz-Mengibar, Sainz-de-Baranda & Santonja-Medina, 2018). First, the inclinometer was placed at the first mark (T1) and it was calibrated to 0°. Then, the inclinometer would be placed on the second mark (T12) and the grades for the thoracic curve would be recorded. After that, the inclinometer was calibrated to 0° again on this mark and then the inclinometer was placed on the third mark (L5–S1) in order to record the lumbar curve angle.

However, the same procedure as in SP was used when it was observed that participants kept their lumbar lordosis in this position.

Maximum flexion of the trunk in a Toe-Touch test position

Firstly, participants were standing on a box 36 cm high with their feet bare and hip-width apart. They were asked to flex the trunk as far as possible, while knees, arms and fingers were fully extended.

The athlete had to keep the MFT for 6–8 s while sagittal spinal curvatures were measured following the same procedure as in the SSP (Sainz de Baranda et al., 2014).

References of normality for thoracic and lumbar curves

The references of normality for thoracic and lumbar curves in each assessed position are described in Table 2.

Table 2 References of normality for thoracic and lumbar curvatures in each position (Ginés-Díaz et al., 2019; Sanz-Mengibar, Sainz-de-Baranda & Santonja-Medina, 2018).

Spinal curve	SP*	SSP*	MFT*	
Category	Ranges	Category	Ranges	Category	Ranges	
Thoracic	Hypokyphosis	<20°	Hypokyphosis	<20°	Hypokyphosis	<40°	
Normal	20–40°	Normal	20–40°	Normal	40–65°	
Hyperkyphosis	>40°	Hyperkyphosis	>40°	Hyperkyphosis	>65°	
Lumbar	Hypolordosis	<20°	Hypokyphosis	<−15°	Hypokyphosis	<10°	
Normal	20–40°	Normal	−15–15°	Normal	10–30°	
Hyperlordosis	>40°	Hyperkyphosis	>15°	Hyperkyphosis	>30°	
Note:

* SP, Standing position; SSP, Slump sitting position; MFT, Maximum flexion of the trunk.

Sagittal integrative morphotype diagnosis

Tables 3 and 4 detail the different categories and subcategories for the integrative diagnosis of the sagittal integrative thoracic and lumbar morphotype, respectively.

Table 3 Classification for thoracic curve’s integrative morphotype diagnosis.

Category	Subcategory	SP*	SSP*	MFT*	
Normal kyphosis		Normal (20–40°)	Normal (20–40°)	Normal (40–65°)	
Functional thoracic hyperkyphois	Static	Normal (20–40°)	Hyperkyphosis (>40°)	Normal (40–65°)	
Dynamic	Normal (20–40°)	Normal (20–40°)	Hyperkyphosis (>65°)	
Total	Normal (20–40°)	Hyperkyphosis (>40°)	Hyperkyphosis (>65°)	
Hyperkyphosis	Total	Hyperkyphosis (>40°)	Hyperkyphosis (>40°)	Hyperkyphosis (>65°)	
Standing	Hyperkyphosis (>40°)	Normal (20–40°)	Normal (40–65°)	
Static	Hyperkyphosis (>40°)	Hyperkyphosis (>40°)	Normal (40–65°)	
Dynamic	Hyperkyphosis (>40°)	Normal (20–40°)	Hyperkyphosis (>65°)	
Hypokyphosis or hypokyphotic attitude	Flat back	Hypokyphosis (<20°)	Hypokyphosis (<20°)	Hypokyphosis (<40°)	
Standing	Hypokyphosis (<20°)	Normal (20–40°)	Normal (40–65°)	
Static	Hypokyphosis (<20°)	Hypokyphosis (<20°)	Normal (40–65°)	
Dynamic	Hypokyphosis (<20°)	Normal (20–40°)	Hypokyphosis (<40°)	
Hypomobile kyphosis		Normal (20–40°)	Normal (20–40°)	Hypokyphosis (<40°)	
Note:

* SP, Standing position; SSP, Slump sitting position; MFT, Maximum flexion of the trunk.

Table 4 Classification for the diagnosis of sagittal integrative lumbar morphotype.

Category	Subcategory	SP*	SSP*	MFT*	
Normal lumbar curve		Normal (20–40°)	Normal (0 ± 15°)	Normal (10–30°)	
Lumbar spine with reduced mobility	Functional lumbar lordosis or hypomobile lordosis	Normal (20–40°)	Normal (0 ± 15°)	Hypokyphosis or lordosis (<10°)	
Lumbar hypomobility	Hypolordosis (<20°)	Normal (0 ± 15°)	Hypokyphosis (<10°)	
Hyperlordotic attitude		Hyperlordosis (>40°)	Normal (0 ± 15°)	Normal (10–30°)	
Functional lumbar hyperkyphosis	Static	Normal (20–40°)	Hyperkyphosis (>15°)	Normal (10–30°)	
Dynamic	Normal (20–40°)	Normal (0 ± 15°)	Hyperkyphosis (>30°)	
Total	Normal (20–40°)	Hyperkyphosis (>15°)	Hyperkyphosis (>30°)	
Lumbar hypermobility	Hypermobility 1	Hyperlordosis (>40°)	Hyperkyphosis (>15°)	Hyperkyphosis (>30°)	
Hypermobility 2	Hyperlordosis (>40°)	Normal (0 ± 15°)	Hyperkyphosis (>30°)	
Hypermobility 3	Hyperlordosis (>40°)	Hyperkyphosis (>15°)	Normal (10–30°)	
Hypolordosis	Hypolordotic attitude	Hypolordosis (<20°)	Normal (0 ± 15°)	Normal (10–30°)	
Lumbar kyphosis 1	Hypolordosis (<20°)	Hyperkyphosis (>15°)	Hyperkyphosis (>30°)	
Lumbar kyphosis 2	Hypolordosis (<20°)	Hyperkyphosis (>15°)	Normal (10–30°)	
Lumbar kyphosis 3	Hypolordosis (<20°)	Normal (0 ± 15°)	Hyperkyphosis (>30°)	
Structured hyperlordosis		Hyperlordosis (>40°)	Hyperlordosis (<−15°) or normal (0 ± 15°)	Lordosis or Hypokyphosis (<10°)	
Structured lumbar kyphosis		Hypolordosis or kyphosis (<20°)	Hyperkyphosis (>15°)	Hyperkyphosis (>30°)	
Note:

* SP, Standing position; SSP, Slump sitting position; MFT, Maximum flexion of the trunk.

Hip joint angle test: L-H fx

The HJA is a field-based test that might be proposed as an alternative to the Passive Straight Leg Raise test or the SRT for the assessment of hamstrings flexibility. The score achieved in this test is not negatively influenced by the pelvic position or stability and only one examiner and an inexpensive gravity goniometer are required (Ayala et al., 2013; Sainz de Baranda et al., 2014). The HJA test can be measured at the end point of maximal trunk flexion in a horizontal or vertical position (Ayala et al., 2013; Sainz de Baranda et al., 2014).

In the current study, the L-H fx was measured with a goniometer while the subject was performing a MFT in a horizontal position (Sainz de Baranda et al., 2014; Santonja, 1996; Santonja, Ferrer & Andújar, 1994). The branches of the goniometer were aligned with the horizontal line and the spinous processes of L4–S2 in order to record the angle between the two references, however, the supplementary angle was used for the data analysis (Fig. 2). Pelvic tilt data were classified as normal (≤100°), mild posterior pelvic tilt (101–110°) and moderate posterior pelvic tilt (>110°).

Figure 2 Hip joint angle test for the measurement of the L-H fx in a maximal flexion of the trunk.

(A) Recorded angle. (B) Supplementary angle.

Statistical analysis

Descriptive statistics including means and standard deviations, minimum and maximum were calculated for each variable: age, weight, height, BMI, training hours per year, training volume, sagittal spinal angles in SP, SSP and MFT and sagittal pelvic disposition.

Prior to the statistical analysis, the distribution of raw data sets was checked using the Kolmogorov–Smirnov test to determine normal distribution. The results demonstrated that the sagittal spinal variables were normally-distributed (p > 0.05). Therefore, parametric analyses were carried out in order to compare sagittal spinal mean angles by competition categories and training workload. Pairwise comparison of means (Student t-test for independent samples) was used to examine the differences between groups of training hours per year and training volume in relation to sagittal spinal and pelvic disposition angles. One-Way Analysis of Variance was carried out to analyze sagittal spinal mean angles between competition categories. Furthermore, the absolute and relative frequency of athletes in each category of spinal morphotype and pelvic disposition were also calculated. Likewise, it was also calculated the absolute and relative frequency of players in each category and subcategory according to their “Sagittal Integrative Morphotype.”

The analysis was performed using SPSS version 23.0 (SPSS Inc., Chicago, IL, USA). The level of significance (α) was set at 0.05; therefore, p values less than 0.05 were considered to be statistically significant.

Results

Sagittal thoracic and lumbar morphotype & pelvic disposition

The means and standard deviations values for spinal curves in each of the three positions and for values of pelvic disposition are shown in Table 5 according to competition categories and training workload.

Table 5 Mean values of spinal curvatures, minimum and maximum of players within each position and for the pelvic disposition*.

Variables	Thoracic curve (degrees)	Lumbar curve (degrees)	Pelvic L-H fx (degrees)	
SP	SSP	MFT	SP	SSP	MFT	
Total (n = 74)	38.5 ± 7.9	45.2 ± 11.2	53.7 ± 10.1	28.7 ± 7.4	20.3 ± 10.4	31.5 ± 8.9	110 ± 10.8	
Categories	
 U11 (n = 30)	38.9 ± 7	44.5 ± 8.5	50.3 ± 9†	29.6 ± 7.7	20.1 ± 10.4	32.3 ± 8.4	114.1 ± 9.3†	
 U13 (n = 25)	38.9 ± 7.6	46 ± 9	54.9 ± 10.5	28.5 ± 6.2	21.3 ± 10.5	32.1 ± 8.9	107.9 ± 10.5	
 U16 (n = 19)	37.2 ± 9.6	45.2 ± 16.8	57.5 ± 9.8†	27.6 ± 8.6	19 ± 9.4	29.3 ± 9.7	106.2 ± 11.8†	
Training h/year	
 <160 h (n = 39)	36.7 ± 8.6*	42 ± 9.9*	51.6 ± 9.6	27.6 ± 8.1	20.1 ± 10.4	33.3 ± 8.2	109.9 ± 11.4	
 >160 h (n = 35)	40.4 ± 6.6*	48.7 ± 11.7*	56 ± 10.2	29.7 ± 6.6	20.4 ± 9.9	29.5 ± 9.3	110.1 ± 10.4	
Training volume	
 <480 h (n = 40)	38.7 ± 7.3	44.4 ± 8.4	52 ± 9.4	28.8 ± 7.4	20.3 ± 10.9	32.6 ± 7.5	110.9 ± 10.7	
 <480 h (n = 40)	38.3 ± 8.6	46.11 ± 13.9	55.7 ± 10.7	28.5 ± 7.5	20.2 ± 9.2	30.1 ± 10.2	109 ± 11.1	
Notes:

SP, Standing position; SSP, Slump sitting position; MFT, Maximum flexion of the trunk; L-H fx, Lumbo-Horizontal angle in flexion.

* Significant differences by training hours per year (p < 0.05).

† Significant differences by categories (p < 0.05).

As it can be observed, there were significant differences across competition categories for the thoracic curve in MFT. Concretely, it was found that U16 had a significant higher dorsal kyphosis than U11 (F(2, 71) = 3.459; p = 0.037; η2 = 0.09) in MFT. In contrast, U11 presented a greater posterior pelvic tilt than U16 (F(2, 71) = 4.082; p = 0.021; η2 = 0.1).

With regard to “training hours per year,” it was found that those who trained >160 h per year had higher dorsal kyphosis in SP (t(72) = −2.051; p = 0.044; d = 0.63) and in SSP (t(72) = −2.694; p = 0.009; d = 0.48) than those who had less training load per year. In addition, it was observed a tendency toward signification for dorsal and lumbar kyphosis in MFT (p = 0.065 and p = 0.067, respectively), presenting a greater dorsal kyphosis and a less pronounced lumbar kyphosis those who trained >160 h per year. Nevertheless, no statistically significant differences were found when spinal curves in each position were analyzed depending on the “training volume.”

Taking into account the references of normality presented in Table 2, it is observed that IH players have an angular mean above normal in SSP for both the dorsal and lumbar curvature.

Table 6 shows the percentage and frequency of athletes within each category by assessment position for each spinal curvature and for pelvic disposition.

Table 6 Percentage and absolute and relative frequency of players within each category by assessment position for each spinal curve and pelvic disposition according to normality references.

Variable	Position*	Category	Mean ± SD	n	%	
Thoracic curve	SP	Rectification (<20°)	16 ± 0.0°	1	1.4	
Normal (20–40°)	34.4 ± 5.5°	45	60.8	
Hyperkyphosis (≥41°)	46 ± 3.8°	28	37.8	
SSP	Hypokyphosis (<20°)	–	0	0	
Normal (20–40°)	33.2 ± 6.4°	26	35.1	
Hyperkyphosis (≥41°)	51.4 ± 7.5°	48	64.9	
MFT	Hypokyphosis (<40°)	36 ± 2.5°	6	8.1	
Normal (40–65°)	52.3 ± 7.1°	55	74.3	
Hyperkyphosis (≥66°)	68 ± 1.8°	13	17.6	
Lumbar curve	SP	Rectification (<20°)	14.9 ± 5.1°	7	9.5	
Normal (20–40°)	29.9 ± 5.9°	66	89.2	
Hyperlordosis (≥41°)	42 ± 0°	1	1.4	
SSP	Hypokyphosis (<−15°)	–	0	0	
Normal (−15–15°)	8.2 ± 4°	23	31.1	
Hyperkyphosis (≥16°)	25.7 ± 6.8°	51	68.9	
MFT	Hypokyphosis (<10°)	–	0	0	
Normal (10–30°)	24.9 ± 5.1°	41	55.4	
Hyperkyphosis (≥31°)	38.8 ± 4.9°	33	44.6	
Pelvic L-H fx	MFT	Normal (≤100°)	94.1 ± 3.8	12	16.2	
Mild posterior pelvic tilt (101–110°)	103.8 ± 2.9	31	41.9	
Moderate posterior pelvic tilt (>110°)	113.5 ± 3.6	31	41.9	
Note:

* SP = Standing position; SSP = Slump sitting position; MFT = Maximum flexion of the trunk; L-H fx = Lumbo-Horizontal angle in flexion.

As for the relaxed SP, the results showed that 60.8% of the athletes presented normal kyphosis, 37.8% had hyperkyphosis, and 1.4% had rectification (hypo- or reduced kyphosis) for the thoracic curve, while 89.2% of the athletes were classified as normal, 1.4% had hyperlordosis and 9.5% presented rectification (hypo- or reduced lordosis) for the lumbar curvature.

With regard to the SSP, the results showed that 35.1% of the athletes presented normal kyphosis, 64.9% had hyperkyphosis, and 1.4% had hypokyphosis for the thoracic curve. On the other hand, 31.1% were within normal ranges, 68.9% had hyperkyphosis and 0% presented hypokyphosis for the lumbar curve.

In a MFT, 74.3% of the athletes presented normal kyphosis, 17.6% had hyperkyphosis, and 8.1% had hypokyphosis for the thoracic curve. As for the lumbar curvature, the results showed that 55.4% had a normal lumbar curve, 44.6% had hyperkyphosis and 0% presented hypokyphosis.

When the L-H fx was evaluated, the results showed that only 16.2% of the athletes were classified as normal, whereas most of IH players were categorized in a posterior pelvic tilt (41.9% with a mild posterior pelvic tilt and 41.9% with a moderate posterior pelvic tilt).

Sagittal integrative spinal morphotype

The values for the sagittal morphotype of the spine integrating the three assessed positions (SP, SSP and MFT) can be observed in Tables 6 and 7. Both tables show the frequency of IH players in each category according to the integrative diagnosis of the sagittal spinal morphotype (Santonja, 1996).

Table 7 Absolute and relative frequency of IH players within each category of thoracic integrative morphotypea.

Category	Subcategory	Classification for integrative thoracic morphotype*	n	%	
SP	SSP	MFT	
Hypokyphosis or hypokyphotic attitude	Standing	Hypokyphosis (<20°)	Normal (20–40°)	Normal (40–65°)	1	1.4	
Hypomobile kyphosis		Normal (20–40°)	Normal (20–40°)	Hypokyphosis (<40°)	1	1.4	
Normal kyphosis		Normal (20–40°)	Normal (20–40°)	Normal (40–65°)	13	17.6	
Hyperkyphosis	Total	Hyperkyphosis (>40°)	Hyperkyphosis (>40°)	Hyperkyphosis (>65°)	12	16.2	
Standing	Hyperkyphosis (>40°)	Normal (20–40°)	Normal (40–65°)	4	5.4	
Static	Hyperkyphosis (>40°)	Hyperkyphosis (>40°)	Normal (40–65°)	9	12.2	
Dynamic	Hyperkyphosis (>40°)	Normal (20–40°)	Hyperkyphosis (>65°)	3	4.1	
Functional hyperkyphosis	Static	Normal (20–40°)	Hyperkyphosis (>40°)	Normal (40–65°)	13	17.6	
Dynamic	Normal (20–40°)	Normal (20–40°)	Hyperkyphosis (>65°)	4	5.4	
Total	Normal (20–40°)	Hyperkyphosis (>40°)	Hyperkyphosis (>65°)	14	18.9	
Notes:

a n: number of cases; %: number of cases with respect to the total IH players.

* Classification of thoracic integrative morphotype according to thoracic values in SP, SSP and in MFT (Santonja, 1996).

With regard to sagittal thoracic morphotype, only 13 IH players presented “Normal Morphotype” with a normal kyphosis in the 3 measurement positions. Thirty-one IH players adopted a normal kyphosis in a relaxed SP, but with an increased kyphosis (hyperkyphosis) in a SSP (static) or in MTF (dynamic), and they were diagnosed with “Functional Thoracic Hyperkyphosis.” Twenty-eight IH players were diagnosed with “Hyperkyphosis” because they adopted a hyperkyphotic curvature in a SP and in a SSP (static) or in MFT (dynamic). When a player presented a hyperkyphotic morphotype in the three positions he was categorized as “Total Hyperkyphosis.” Only one IH player was diagnosed with “Hypomobile Kyphosis” (adopted a normal kyphosis in a relaxed SP and in a SSP, but presented a hypokyphosis in MFT), and another player with “Hypokyphosis or Hypokyphotic Attitude” (adopted a normal kyphosis in a SSP and in MFT, while a hypokyphosis is presented in a relaxed SP) (Table 7).

With regard to the sagittal integrative lumbar morphotype (Table 8), only 17 IH players presented “Normal morphotype” with a normal lumbar curvature in the three assessed positions. Forty-nine IH players adopted a normal kyphosis in a relaxed SP, but with an increased kyphosis (hyperkyphosis) in a SSP (static) (n = 15) or in a MFT (dynamic) (n = 4), or in both positions (total) (n = 30), and they were diagnosed with “Functional Lumbar Hyperkyphosis.” Five IH players were diagnosed with “Structured Lumbar Kyphosis” because they presented a hypolordosis or kyphosis in a SP and a hyperkyphosis in a SSP and in MFT. Only two IH players were diagnosed with “Hypolordosis” (with a hypolordosis in a relaxed SP, but a normal lordosis in a SSP and in MFT). Finally, another IH player was diagnosed with “Lumbar Hypermobility.” No players presented the morphotype “Hyperlordotic Attitude” or “Structured Hyperlordosis.”

Table 8 Absolute and relative frequency of IH players within each category of integrative lumbar morphotypea.

Category	Subcategory	Classification for integrative lumbar morphotype*	n	%	
SP	SSP	MFT	
Hypolordosis	Lumbar hypomobility	Hypolordotic attitude (<20°)	Normal (0 ± 15°)	Normal (10–30°)	2	2.7	
Normal lumbar curve		Normal (20–40°)	Normal (0 ± 15°)	Normal (10–30°)	17	23	
Functional lumbar hyperkyphosis	Static	Normal (20–40°)	Hyperkyphosis (>15°)	Normal (10–30°)	15	20.3	
Dynamic	Normal (20–40°)	Normal (0 ± 15°)	Hyperkyphosis (>30°)	4	5.4	
Total	Normal (20–40°)	Hyperkyphosis (>15°)	Hyperkyphosis (10–30°)	30	40.5	
Lumbar hypermobility		Hyperlordosis (>40°)	Normal (0 ± 15°) or Hyperkyphosis (>15°)	Normal (10–30°) or Hyperkyphosis (>30°)	1	1.4	
Structured lumbar kyphosis		Hypolordosis or kyphosis (<20°)	Hyperkyphosis (>15°)	Hyperkyphosis (>30°)	5	6.8	
Notes:

a n, number of cases; %, number of cases with respect to the total IH players.

* Classification of integrative thoracic morphotype according to thoracic values in a SP, in a SSP and in MFT (Santonja, 1996).

Discussion

This study was undertaken to investigate the sagittal spinal curvatures of the thoracic and lumbar spine and its relationship to training load, and to describe the “Sagittal Integrative Morphotype” in young federated IH players.

IH has specific requirements for players, which include trunk forward bending and concrete movements such as skating, generating physical needs that do not occur in other sports. These physical demands make IH players susceptible to certain postural and structural adaptations that can result in subsequent injuries and back pain. Previous studies have shown that those specific and repetitive movements and postures of each sport influence spinal curvatures (Rajabi et al., 2008; Uetake et al., 1998; Wodecki et al., 2002) and for that reason, several studies agree on the importance of a initial postural evaluation in order to identify spinal deformities and sagittal imbalances. Sagittal curvatures are geometric parameters which influence mechanical properties of the spine during compressive loading (Harrison et al., 2005; Keller et al., 2005). Sagittal alignment influences postural loading and the load balance of the intervertebral disc, therefore, abnormal spinal curvatures cause increased forces to act upon the intervertebral discs (Keller et al., 2005). Alterations in spinal curvatures may potentially influence the development of lower back pain (Harrison et al., 2005; Smith, O’Sullivan & Straker, 2008), which is a common pathology among athletes (Kameyama et al., 1995).

The most reliable technique to quantify kyphosis and lordosis is the conventional spinal X-ray method. There are other methods free of ionizing radiation that assess the curvatures of the spine in the sagittal plane, for instance, the inclinometer provides a noninvasive evaluation with good reproducibility, reliability and correlation with the radiographic measurement (López-Miñarro et al., 2008; Sainz de Baranda, Rodríguez-García & Santonja, 2010; Sainz de Baranda, Santonja & Rodríguez-Iniesta, 2009; Sanz-Mengibar, Sainz-de-Baranda & Santonja-Medina, 2018).

The integrative diagnosis of the sagittal morphotype of the spine was defined by Santonja (1996) and adds the assessment of the sagittal curvatures during MFT and in a SSP (Sainz de Baranda, Rodríguez-García & Santonja, 2010; Sainz de Baranda, Santonja & Rodríguez-Iniesta, 2009; Santonja, 1996; Sanz-Mengibar, Sainz-de-Baranda & Santonja-Medina, 2018) to the classical quantification of the thoracic and lumbar curves in a relaxed SP in order to perform a more accurate diagnosis.

Reference values and categories for thoracic curvature in previous studies

In the current study, mean thoracic curvature value was 38.5°, 45.2° and 53.7° in a relaxed SP, in a SSP and in MFT, respectively.

Wojtys et al. (2000) found similar values (a mean of 38.1°) when they studied the thoracic curve in 189 ice-hockey players (aged between 8 and 18 years) in a relaxed SP. In sports like basketball, handball, volleyball and female artistic gymnast, some studies have found similar or lower angular values; while other studies found higher values in swimmers, runners, tennis players, trampoline gymnasts, male artistic gymnasts, cross-country skiers, and paddlers (Table 9). On the other hand, it is interesting to note that in sports related to dancing abilities, thoracic angular values tend to be much lower than in other types of sports (Gómez-Lozano, 2007; Gómez-Lozano et al., 2013; Nilsson, Wykman & Leanderson, 1993).

Table 9 Angular values for thoracic curvature in a relaxed standing position, in a slump sitting position and in maximal trunk flexion in different previous studies.

	Present study (2019)	(Wojtys et al., 2000)	(Rajabi, Alizadeh & Mobarakabadi, 2007)	(Rajabi et al., 2012)	(Alricsson et al., 2016)	(Pastor et al., 2002)	(López-Miñarro, Alacid & Muyor, 2009)	(López-Miñarro et al., 2008)	(Sainz de Baranda, Santonja & Rodríguez-Iniesta, 2009)	(Sanz-Mengibar, Sainz-de-Baranda & Santonja-Medina, 2018)	
Aged (years)	8–15	8–18	15–34	18–19	16–19	9–15	13–14	13.3	14.9	15.02	
Sports	Inline hockey	Ice-hockey	Field hockey	Field hockey	Cross-Country	Swimming	Running	Paddlers	Trampoline gymnasts	Artistic gymnasts	
SP	38.5°	38.1°	34.1°	Athletes: 41.71°	41.2°	♂: 40.4°	45.6°	Kayak: 42.2°	♂: 46.9°	♂: 39.6°	
Non-athletes: 36.72°	♀: 39.5°	Canoe: 37.4°	♀: 43°	♀: 31.8°	
SSP	45°	–	–	–	–	–	–	Kayak/canoe: ~50°	♂: 51.3°	♂: 39.6°	
♀: 49.2°	♀: 31.8°	
MFT	53.7°		–	–	–	♂: 78.45°	63.5°	Kayak/canoe: ~65°	♂: 55.7°	♂: 55.5°	
♀: 73.4°	♀: 47.4°	♀: 49.3°	
	(Sainz de Baranda, Rodríguez-García & Santonja, 2010)	(Ferreira-Guedes & Amado-João, 2014)	(Grabara, 2016a)	(Grabara, 2012)	(Grabara, 2015)	(Grabara & Hadzik, 2009)	(Muyor et al., 2013)	(Grabara, 2014a)	(Grabara, 2014b)		
Aged (years)	15	12–16	13	13–15	14–16	13–16	13–18	12–15	14–17		
Sports	Trampoline gymnasts	Basketball	Basketball	Basketball	Volleyball	Volleyball	Tennis players	Handball	Volleyball Basketball Handball		
SP	Training’s h/ys	30.4°	1-yes: 38.5°	13–14 yr: 28.8°	14 yr: 30.1°	13–14 yr: 27.2°	♂: 43.8°	12 yr: 27.5°	Vb ♂: 39.6°/ ♀: 38.2°		
≤2000 h: 43.9°	2-yes: 35.8°	15 yr: 27.2°	15 yr: 31.1°	15–16 yr: 29.6°	♀: 36.1°	13 yr: 27.2°	Hb ♂: 35.9°		
>2000 h: 43.9°	3-yes: 34.4°		16 yr: 30.2°			14 yr: 28.4°	Bb ♂: 34.4°/ ♀: 33.6°		
						15 yr: 28.8°			
SSP	≤2000 h: 52.4°	–	–		–	–	–	–	–		
>2000 h: 48.9°	
MFTT	≤2000 h: 50.6°	–	–		–	–	–	–	–		
>2000 h: 51.2°	

In the current study, a mean angular value of 45.2° was observed in the SSP for the thoracic curve. This mean value is lower than those observed in trampoline gymnasts (Sainz de Baranda, Santonja & Rodríguez-Iniesta, 2009; Sainz de Baranda, Rodríguez-García & Santonja, 2010) and paddlers (López-Miñarro et al., 2008). In contrast, this angular value is higher than those observed in artistic gymnasts (Sanz-Mengibar, Sainz-de-Baranda & Santonja-Medina, 2018).

In the MFT, a mean thoracic angular value of 53.7° was observed among the IH players studied in the present investigation. However, previous research has found higher values in runners, paddlers and male artistic gymnasts (López-Miñarro et al., 2008; López-Miñarro, Alacid & Muyor, 2009). On the other hand, similar or lower angular values for the thoracic curve were observed in trampoline gymnasts and female artistic gymnasts (Sainz de Baranda, Santonja & Rodríguez-Iniesta, 2009; Sanz-Mengibar, Sainz-de-Baranda & Santonja-Medina, 2018).

When the results were analyzed by “training hours per year,” it was found that players who trained more than 160 h per year presented a significantly higher dorsal kyphosis in SP and SSP and a tendency toward signification in MFT than those who had less training load. Furthermore, it was observed that U16 had a significant higher dorsal kyphosis than U11 in MFT. In this sense, varies studies have found a relationship between sports training and variations in the sagittal spinal curvatures of adolescent athletes (Ferreira-Guedes & Amado-João, 2014; Grabara, 2015; Hecimovich & Stomski, 2016; Sainz de Baranda, Santonja & Rodriguez-Iniesta, 2010). Wojtys et al. (2000) reported an increase in sagittal spinal curvature in adolescents who participated in ice hockey and exceeded 400 h of training per year. Intense physical training combined with a developing spine, where loads are transferred from the upper to the lower extremities, leads to an overload of the spinal structures that could influence the deformation of the spine.

In sports with predominance of the trunk forward bending position, like IH, skiing, canoeing, cycling or show jumping riding, it has been found a high percentage of thoracic hyperkyphotic postures (Alricsson & Werner, 2006; Förster et al., 2009; Ginés-Díaz et al., 2019; López-Miñarro et al., 2008; Rajabi et al., 2008). In contrast, repetitive back hyperextension movements or a remain trunk extension position, which are popular among rhythmic gymnasts or classical dancers tend to flatten the normal thoracic curve to a thoracic hypokyphosis (Grabara, 2015; Kums et al., 2007; Nilsson, Wykman & Leanderson, 1993) as well as to increase the lumbar curve and generate lumbar hyperlordosis (Falter & Hellerer, 1982).

Inline hockey is a very fast paced game, which is characterized by high intensity intermittent skating, rapid changes in velocity and duration, frequent body contact, and the execution of a wide variety of technical skills (Flik, Lyman & Marx, 2005; Mölsä et al., 2003). IH implicates adapting the body to a hard physical effort and to the required posture for that sport. As a result, athletes commonly present postures that are related to the most common sports abilities in each discipline (Rajabi et al., 2008; Usabiaga et al., 1997). As Wojtys et al. (2000) stated, specific postures and actions which take place in IH practice might modify the sagittal spinal curvatures by altering spine’s exposure to certain mechanical loads during the athlete’s growth. In adolescents, Ferreira-Guedes & Amado-João (2014) note that “these biomechanical compensations may influence the growth processes and lead to the development of various postural patterns due to the immaturity of their musculoskeletal structures. At first, the postural compensations are adaptive, but later they can become permanent and even predispose young athletes to injuries.” So, the results of the current study could confirm that sagittal curvatures of the spine can be modified with regular IH training as previously described in other sports (Ginés-Díaz et al., 2019; Sainz de Baranda, Santonja & Rodríguez-Iniesta, 2009; Sanz-Mengibar, Sainz-de-Baranda & Santonja-Medina, 2018; Uetake et al., 1998).

In the current study, most of IH players had normal angular values in a relaxed SP (n = 45/74, 60.8%) and in a MFT (n = 55/74, 74.3%) for the thoracic curve (Fig. 3). However, in the SSP there was a higher percentage of IH players with an increased thoracic curvature or hyperkyphosis (n = 48/74, 64.9%).

Figure 3 Frequency and percentage of IH players by category of thoracic curve in each of the three positions.

The percentages of normality in a relaxed SP have been greater than in previous studies. For instance, Grabara (2016a) found 60–70% of hyperkyphosis in 10 basketball players who were 13 years old. Likewise, Pastor et al. (2002) found 57.1% and 46.5% of male and female young elite swimmers with hyperkyphosis. López-Miñarro, Alacid & Muyor (2009) found 37% of young kayakers having neutral thoracic kyphosis and 63% with hyperkyphosis, while Muyor, López-Miñarro & Alacid (2011a) reported that 41.7% of elite cyclists showed neutral thoracic kyphosis and 58.3% presented thoracic hyperkyphosis. In another study, Muyor, López-Miñarro & Alacid (2011b) reported that elite cyclists showed a statistically higher thoracic hyperkyphosis than non-athlete subjects. These authors justified their findings with specific sport adaptations. In this sense, Grabara & Hadzik (2009) found that a kyphotic posture tended to be more frequent and the lordotic one less frequent in volleyball players than in untrained subjects. The authors attributed that finding to the typical volleyball posture consisting of forwarding bending with rounded back as well as the arms and shoulders protruding. Wojtys et al. (2000) reported that high-intensity training increases the risk of developing adolescent hyperkyphosis. In this sense, Alricsson & Werner (2006) found that after 5 years of intensive training the skiers increased their thoracic kyphosis but no change in lumbar lordosis was noticed.

Other studies found lower percentages of thoracic hyperkyphosis. For instance, Muyor et al. (2013) found 37.5% and 6.2% of thoracic hyperkyphosis in 24 male and 16 female elite adolescent tennis players, respectively. López-Miñarro et al. (2008) found a 26.1% and 15% of thoracic hyperkyphosis in 23 kayak paddlers and 20 canoe young athletes, respectively. Finally, Sanz-Mengibar, Sainz-de-Baranda & Santonja-Medina (2018) found 16.6% of thoracic hyperkyphosis in 47 artistic gymnastics who competed in national and international tournaments.

As for the assessment of the spinal curvatures in other positions, Sanz-Mengibar, Sainz-de-Baranda & Santonja-Medina (2018) found a 37% and 79.1% of thoracic hyperkyphosis in artistic gymnastics in a MFT as well as in a SSP, respectively. López-Miñarro et al. (2008) found higher percentages of thoracic hyperkyphosis in infantile male paddlers. The results showed that 25% and 45% of kayak and canoe athletes, respectively, had thoracic hyperkyphosis in MFT. In the same study, these authors found that 82% and 95% of kayak and canoe athletes, respectively, had thoracic hyperkyphosis in a SSP.

Pastor et al. (2002) observed only 24.7% of the morphotypes within normality in swimmers, 29.4% of the morphotypes with mild kyphosis and 45.9% with moderate kyphosis. The same author performed a radiological study in the position of Sit and Reach test (SRT) and observed a higher percentage of moderate and marked thoracic curves (p < 0.05) and a significant tendency to increase the number of vertebral wedges as the value of kyphosis and age increased. In addition, these wedges were related to the dynamic thoracic kyphosis, since the swimmers with more thoraco-lumbar wedges presented higher values of dynamic thoracic kyphosis (p < 0.05).

In contrast, Gómez-Lozano (2007) only observed 6.1% and 3% of misaligned morphotypes in classic and Spanish dancers, respectively, as only some mild hyperkyphotic attitudes were diagnosed in this posture.

It must be pointed out that not all studies use the same spinal assessment protocol and the same references to categorize sagittal spinal angular values. In this sense, Grabara (2016a) established that values above 35° are considered thoracic hyperkyphosis, thoracic normality is considered from 25° to 35° and thoracic hypokyphosis is accepted when the value is lower than 25°. Muyor et al. (2013) used the references of normality proposed by Mejia et al. (1996) and Tüzün et al. (1999), where the values between 20° and 45° are accepted as neutral thoracic kyphosis, values below 20° are considered thoracic hypokyphosis, and values above 45° are considered thoracic hyperkyphosis. However, Pastor et al. (2002) and Gómez-Lozano (2007) used the same reference values as in the current study.

Reference values and categories of lumbar curvature

Regarding the lumbar curvature, in the current study mean values were 28.7°, 20.3° and 31.5° in a relaxed SP, in a SSP and in MFT, respectively.

In comparison with our results, Wojtys et al. (2000) found an average value much higher (44.5°) for the lumbar curvature in a relaxed SP in 189 ice-hockey players aged from 8 to 18 years old. In all the sports participants whose spinal morphotype has been assessed (Table 10), higher values have been observed in this position, except in basketball players (Ferreira-Guedes & Amado-João, 2014) and in artistic gymnasts (Sanz-Mengibar, Sainz-de-Baranda & Santonja-Medina, 2018).

Table 10 Angular values for lumbar curvature in a relaxed SP, in a SSP and in MFT in by sport.

	Present study (2019)	(Wojtys et al., 2000)	(Kujala et al., 1997)	(Ogurkowska & Kawałek, 2017)	(Alricsson et al., 2016)	(Grabara, 2012)	(Ferreira-Guedes & Amado-João, 2014)	(Grabara, 2016b)	(Grabara, 2014a)	
Aged (years)	8–15	8–18	11.9	24–35	16–19	13–15	12–16	13	12–15	
Sports	Inline hockey	Ice hockey	Ice hockey	Field hockey	Cross-country skiers	Basketball	Basketball	Basketball	Handball	
SP	28.7°	44.5°	35°	43.2	33.4°	13–14 yr: 27.6°	32.8°	1-yes: 21.5°	12 yr: 30.7°	
15 yr: 27.8°	2-yes: 29°	13 yr: 28.6°	
	3-yes: 24.6°	14 yr: 28.1°	
		15 yr: 25.9°	
SSP	20.3°	–	–	–	–	–	–	–	–	
MFT	31.5°	–	–	–	–	–	–	–	–	
	(Grabara, 2015)	(Grabara & Hadzik, 2009)	(Pastor et al., 2002)	(López-Miñarro, Alacid & Muyor, 2009)	(Sainz de Baranda, Santonja & Rodríguez-Iniesta, 2009)	(Sainz de Baranda, Rodríguez-García & Santonja, 2010)	(Sanz-Mengibar, Sainz-de-Baranda & Santonja-Medina, 2018)	(López-Miñarro et al., 2008)	(Gómez-Lozano, 2007)	
Aged (years)	14–16	13–16	9–15	13–14	14.9	15	15.02	13.3		
Sports	Volleyball	Volleyball	Swimming	Running	Trampoline gymnasts	Trampoline gymnasts	Artistic gymnasts	Kayak Canoe	Dancers	
SP	14 yr: 30.1°	13–14 yr: 28°	♂: 31.21°	31.2°	♂: 32°	Training’s hours/ys	♂: 39.6°	Kayak: 27.9°	♀: 35.18°	
15 yr: 31.1°	15–16 yr: 25.5°	♀: 36.33°	♀: 40.3°	≤2000 h: 31.7°	♀: 30.5°	Canoe: 25.7°	♀: 33.84°	
16 yr: 30.2°				>2000 h: 36.6°				
SSP	–	–		–	♂: 21°	≤2000 h: 21°	♂: 26.1°	Kayak/canoe: ~18°	♀: 8.33°	
♀: 14°	>2000 h: 16.4°	♀: 27.7°	♀: 8.36°	
MFT	–	–	♂: 24,62°	27.4°	♂: 31.9°	≤2000 h: 32.3°	♂: 15.5°	Kayak/canoe: ~30°	♀: 19.82°	
♀: 21°	♀: 26.7°	>2000 h: 27.5°	♀: 15.7°	♀: 19.48°	

Sainz de Baranda, Santonja & Rodríguez-Iniesta (2009) found a mean value of 36.25 ± 10.1° with 69 competition gymnasts of the Trampoline modality (35 girls and 34 boys). When the results were compared by sex, it was observed a greater lumbar lordosis in girls (40.31° ± 10°) than in boys (32.06° ± 7.7°).

When Ohlén, Wredmark & Spangfort (1989) assessed the spinal morphotype, it was found a mean value of 35.6° ± 7.8° for the lumbar curve with a Debrunner’s cifometer and value of 35.2° ± 6.9° with an inclinometer in 64 artistic gymnasts. 20% of the gymnasts manifested lower back pain. When values for the lumbar curve were compared between the gymnasts with pain (40.6° ± 7.9°) and the asymptomatic gymnasts (35.4° ± 7.2°), it was observed that the mean lordotic value was higher in gymnasts with back pain. In addition, the authors found a significant correlation between back pain and a lumbar lordosis greater than 41°.

Martínez-Gallego (2004) observed mean values of 35.88° ± 8.69° in 82 competitive rhythmic gymnasts and values of 40.30° ± 898° in 81 recreational rhythmics gymnasts.

Conesa Ros (2015) observed mean value of 32.9° ± 8.5° in a group of competitive esthetic gymnasts. In addition, the author observed that a lumbar lordosis tended to increase with age. Thus, the group of competitive esthetic gymnasts under 11 years old had a lumbar value of 28° ± 6.8° and the group over 15 years old had a mean lumbar value of 36.4° ± 9.2°. The group of competitive rhythmic gymnasts under 11 had a mean lumbar lordosis of 33.8° ± 9.4°, while the group over 15 years old had a mean lumbar value of 39.2° ± 8.6°.

This evolution of lordosis with age has also been found in previous studies carried out with school-aged children (Cil et al., 2005; Murray & Bulstrode, 1996; Voutsinas & MacEwen, 1986).

With regard to the lumbar curvature in a SSP, the results of the current study showed a mean value of 20.3°. There are few studies which have assessed the sagittal spinal curvatures in a SSP (Conesa Ros, 2015; Gómez-Lozano, 2007; López-Miñarro et al., 2008; Martínez-Gallego, 2004; Sainz de Baranda, Santonja & Rodríguez-Iniesta, 2009; Sanz-Mengibar, Sainz-de-Baranda & Santonja-Medina, 2018).

Sainz de Baranda, Santonja & Rodríguez-Iniesta (2009) observed a mean value of 17.4° ± 9.6° for the lumbar curve in gymnasts of the Trampoline modality. When the results were compared by sex, a significantly greater lumbar kyphosis was observed in males (21° ± 7.9°) than in female gymnasts (14° ± 10°) (p < 0.004).

Conesa Ros (2015) and Martínez-Gallego (2004) showed how sports practice can influence or can be related to a higher angular value for the lumbar curve in their studies with esthetic and rhythmic gymnasts. In this sense, both rhythmic and esthetic gymnasts (16.7° ± 6.6° and 15.9° ± 8.1°, respectively) had a significantly greater lumbar kyphosis than the control group (13.8° ± 7.7°) (p = 0.033).

In the same way, Martínez-Gallego (2004) also observed a greater lumbar kyphosis in a SSP in the rhythmic gymnast’s groups, either recreational (16.24° ± 7.29°) or competitive (168° ± 6.55°), when compared with the control group (13.81° ± 7.72°).

The incorrect alignment of the lumbar spine found in the SSP in the three modalities of gymnastics could be due to repetitive hyperflexions and hyperextensions of the trunk which are performed in gymnastics. Thus, these movements could come to a hypermobile lumbar curve.

López-Miñarro et al. (2008) found angular lumbar values lower than 20° in 43 infantile paddlers (23 kayakers and 20 canoeists), and no significant differences were found between kayakers and canoeists. Likewise, when López-Miñarro et al. (2008, 2013) assessed the sagittal spinal curves of 130 canoeists (aged from 15 to 20 years old), the authors found values lower than 20° for the lumbar curvature, with no significant differences regarding gender.

Sanz-Mengibar, Sainz-de-Baranda & Santonja-Medina (2018) observed mean value of 15.62° ± 6.41° for the lumbar curve in their study with gymnasts of the artistic modality, and no significant differences between boys (15.52° ± 6.92°) and girls (15.71° ± 6.02°) were found. However, Gómez-Lozano (2007) observed a lower mean value of lumbar kyphosis among classic dancers (8.33° ± 6.44°) in a SSP.

With regard to the lumbar curvature in the MFT, the results of the current study showed a mean value of 31.5°. In trampoline gymnasts, runners and paddlers were found similar or lower values (Sainz de Baranda, Rodríguez-García & Santonja, 2010; López-Miñarro, Alacid & Muyor, 2009; Sainz de Baranda, Santonja & Rodríguez-Iniesta, 2009; Sanz-Mengibar, Sainz-de-Baranda & Santonja-Medina, 2018).

Figure 4 indicates that most of the athletes had normal angular values for the lumbar curvature in a relaxed SP (n = 66/74, 89.2%) and in a MFT (n = 41/74, 55.4%). However, there is a higher percentage of IH players with increased angular values or hyperkyphosis (n = 51/74, 68.9%) in a SSP.

Figure 4 Frequency and percentage of IH players by category of lumbar curvature according to normality references in each position.

In the current study, it was found a 9.5% of lumbar rectification and 1.4% of lumbar hyperlordosis in the relaxed SP. In contrast, Pastor et al. (2002) found higher percentages of lumbar hyperlordosis in young elite swimmers (7.1% in males and 32.3% in females). López-Miñarro et al. (2008) reported that 8.7% of 23 kayakers and 10% of 178 canoeists had lumbar rectification. Grabara (2016b) found 50% of hypolordosis and 10% of hyperlordosis in 10 basketball players aged from 13 years old. Recently, Sanz-Mengibar, Sainz-de-Baranda & Santonja-Medina (2018) observed lumbar hyperlordosis in 12.5% of 47 artistic gymnastics.

It was also found that 68.9% and 44.6% of the IH players had lumbar hyperkyphosis in a SSP and in a MFT, respectively.

Some previous studies found higher percentages of hyperkyphosis for the lumbar curvature in these positions, possibly due to the practice of the sport in a sitting position or the repetition of technical gestures with a maximal ROM in the lumbar spine and lower limb. In this sense, López-Miñarro et al. (2008) reported that around 90% of paddlers had lumbar hyperkyphosis in MFT. In addition, these authors found lumbar hyperkyphosis in around 75% of kayak and canoe athletes in a SSP.

In contrast, Sanz-Mengibar, Sainz-de-Baranda & Santonja-Medina (2018) found 39% of lumbar hyperkyphosis in both MTF and in a SSP among artistic gymnasts.

It must be pointed out that not all studies use the same spinal assessment protocol and the same references to categorize sagittal spinal angular values. In this sense, for the relaxed SP, Grabara (2016b) established that values above 35° are considered lumbar hyperlordosis, a neutral lumbar spine is considered from 25° to 35° and lumbar hypolordosis is accepted when the value is lower than 25°. Muyor et al. (2013) used the references of normality proposed by Tüzün et al. (1999), where the values between 20° and 40° are accepted as a neutral lumbar spine, values below 20° are considered as an hypolordotic lumbar spine, and values above 40° are considered hyperlordosis.

With respect to the MFT, Pastor et al. (2002) established that values below 22° are considered as normal lumbar kyphosis and values between 22° and 29° are considered as lumbar hyperkyphosis. As for the SSP, the author established that values below 14° were accepted as normal lumbar kyphosis and values between 14° and 21° were considered lumbar hyperkyphosis.

Pelvic disposition

The flexibility of hamstring muscles is important for the prevention of muscular and postural imbalances, for the maintenance of the full range of motion in the hip flexion as well as for the optimal musculoskeletal function (Sainz de Baranda et al., 2014).

Hamstrings extensibility influences pelvic posture (Congdon, Bohannon & Tiberio, 2005) and spinal curvatures (López-Miñarro, Alacid & Muyor, 2009). Decreased extensibility of hamstring muscles has been associated with a greater thoracic kyphosis and a higher posterior pelvic tilt when maximal trunk flexion with knees extended is performed. Consequently, an incorrect hamstrings extensibility and the constant repetition of trunk hyperflexion and hyperextension due to the sports practice could increase intervertebral stress (Beach et al., 2005) as well as thoracic and lumbar intradiscal pressure (Polga et al., 2004; Wilke et al., 1999), predisposing subjects to spinal disorders (McGill, 2002).

The results of the current study suggest that a hamstring-specific extensibility program is necessary for this group of IH players, especially among the youngest (U11) since they showed a greater posterior pelvic tilt than U16. Only 16.2% of the IH players showed normal values for the L-H fx angle.

In fact, a high percentage of IH players showed decreased hamstrings flexibility, since 41.9% of IH players presented a mild and a moderate posterior pelvic tilt. This lack of flexibility may influence pelvic and spinal postures in a MFT, which is a very common position adopted in the IH techniques. Thus, prior evaluation of hamstring flexibility is recommended to point out specific and individualized preventive programs in order to prevent spinal problems.

Sagittal integrative spinal morphotype

This is the first study in which three different positions are combined for the diagnosis of the sagittal spine in young IH players. There are few studies that perform the “Sagittal Integrative Morphotype” assessment (Collazo, 2015; Ginés-Díaz et al., 2019; Sanz-Mengibar, Sainz-de-Baranda & Santonja-Medina, 2018), so the comparison with existing literature is difficult.

On the one hand, for the thoracic curve, the most frequent diagnosis is “Functional Hypekyphosis” (41.9%), which means that they would mainly need to improve their spinal alignment in a SSP and in MFT. Concretely, most of them presented a “Static Functional Hyperkyphosis” (17.6%) or a “Total Functional Hyperkyphosis” (18.9%). On the other hand, 28 IH players (37.9%) were diagnosed with “Hyperkyphosis.” To be more specific, 16.2% of them had “Total Hyperkyphosis” and 12.2% presented “Static Hyperkyphosis.” These diagnoses are in line with those found in young riders (mean age: 14.55 years), artistic gymnasts (mean age: 15.02 years) and schoolchildren (mean age: 10.55 years) in which the most common thoracic sagittal integral morphotype was “Functional Thoracic Hyperkyphosis” (40%, 62.5% and 36.8%, respectively) (Collazo, 2015; Ginés-Díaz et al., 2019; Sanz-Mengibar, Sainz-de-Baranda & Santonja-Medina, 2018). Since 29.8% of IH players were diagnosed with “Static Hyperkyphosis” and “Static Functional Fyperkyphosis,” these players would also need to improve their posture in a SSP and in the relaxed SP, in fact, these results might be associated with poor postural hygiene while sitting.

As for the lumbar curvature, most of IH players (n = 49; 66.2%) were diagnosed with “Functional Lumbar Hyperkyphosis.” Specifically, 20.3% of players had “Static Functional Lumbar Hyperkyphosis” and 40.5% presented a “Total Functional Lumbar Hyperkyphosis.” Again, these results coincide with previous studies (riders: 40%, artistic gymnasts: 39.58%, schoolchildren: 82.4%) (Collazo, 2015; Ginés-Díaz et al., 2019; Sanz-Mengibar, Sainz-de-Baranda & Santonja-Medina, 2018).

In this sense, as Purcell & Micheli (2009) stated, repetitive flexo-extension and torsion movements because of technical-tactical actions in IH can result in overuse injuries to the spine. In fact, these repetitive movements with an imbalanced spinal posture (e.g., hyperkyphotic position with the trunk bent forward) are particularly worrisome in young IH players. Therefore, the position of the lumbar spine while sitting or trunk forward bending should be trained for a better alignment through pelvic proprioceptive exercises or trunk muscles strengthening.

Furthermore, it is important to highlight the fact that 6.8% of the IH players were diagnosed with “Structured Lumbar Kyphosis.” In this sense, it has to be pointed out that the high prevalence of posterior pelvic tilt found among players could have led to a misaligned lumbar spine during MFT probably due to the hamstrings tightness that can make the pelvis lose its horizontality and adopt a posterior pelvic tilt (López-Miñarro et al., 2012; Santonja, Andújar & González-Moro, 1994). Since the trunk forward bending while standing is the basic posture in IH, the players should train their hamstrings flexibility in order to keep a neutral lumbar spine when their trunk is bent forward. In addition, good pelvic proprioception would be necessary to keep a neutral pelvic tilt while IH players have to stay in constant quadruple flexion (ankle, knee, hip and trunk). It is important to note that if sagittal spinal assessment had been only carried out in SP, the results of this study would have shown that most of IH players were within the normal ranges for both curves (60.8% of the athletes presented normal kyphosis and the 89.2% of the athletes presented normal lordosis in the SP). However, taking into account the “Sagittal Integrative Morphotype” it was determined that IH players had a misaligned sagittal spine. These results show how important is to include the assessment of the three positions as part of the protocol in order to define “Sagittal Integrative Morphotype” and so as to establish a correct diagnostic (Sanz-Mengibar, Sainz-de-Baranda & Santonja-Medina, 2018). Therefore, an incorrect sagittal spinal assessment leads to misclassification of the athletes’ morphotypes, generating negative consequences, not only in terms of deformity and pain but also in preventive and rehabilitative terms (Ginés-Díaz et al., 2019).

Some limitations of the present study must be reported. The age distribution of the participants was relatively limited and the sample size was relatively small. Furthermore, only male IH players were assessed. In addition, patients with radiographic evidence deformity may present a normal spinal morphotype. Future studies which include a larger sample should investigate the association between sagittal spinal morphotype and back pain or the ratio of injury. Furthermore, prospective investigations in order to study how sagittal spinal curves develop with age and practice are needed.

Conclusions

Federative IH practice seems to cause specific adaptations in spinal sagittal morphotype in young players. The findings reported in the present study suggest an association between exposure to IH athletic training, age and increased thoracic kyphosis.

The most prevalent sagittal spinal misalignments in young federated male IH players were the thoracic hyperkyphosis (64.9%) and the lumbar hyperkyphosis (68.9%) in a SSP and the lumbar hyperkyphosis (44.6%) in a MFT.

However, taking into account the “Sagittal Integrative Morphotype” only 13 (17.6%) IH players presented “Normal Morphotype” with a normal thoracic kyphosis in the three measured positions. While only 17 (23%) IH players presented “Normal Lumbar Morphotype” with a normal lumbar curvature in the three assessed positions.

For the thoracic curvature, 18.9% of the IH players presented “Total Functional Hyperkyphosis,” 17.6% of the players presented a “Static Functional Hyperkyphosis” and 16.2% of the players had a “Total Hyperkyphosis”. Whereas for the lumbar curvature a 40.5% of the players presented “Total Functional Hyperkyphosis” and the 20.3% of players were diagnosed with “Static Functional Hyperkyphosis”. Furthermore, only 16.2% of IH players showed normal pelvic titl.

Clinical implications

It is important to assess the “Sagittal Integrative Spinal Morphotype” in sports for the preemptive care of spinal deformities from the earliest stages. The assessment of sagittal spinal curvatures in the three described positions gives a new perspective for the diagnosis of sagittal spinal morphotype, which may reduce the number of players wrongly classified as normal. It is important to note that the protocol is easy to apply due to the low-cost instruments used and it only requires an examiner.

Exercise programs to prevent or rehabilitate these imbalances in young IH players are needed. Pelvic proprioceptive exercises, trunk muscles strengthening, and flexibility training could be included as a part of preventive programs. This manuscript creates a paradigm for future studies about associated risk factors to develop unbalanced sagittal spines in IH players.

Supplemental Information

Supplemental Information 1 The raw measurements.

Click here for additional data file.

Supplemental Information 2 Codebook.

Click here for additional data file.

Additional Information and Declarations

Competing Interests

Author Contributions

Human Ethics

Data Availability

The authors declare that they have no competing interests.

Pilar Sainz de Baranda conceived and designed the experiments, prepared figures and/or tables, authored or reviewed drafts of the paper, approved the final draft.

Antonio Cejudo conceived and designed the experiments, performed the experiments, analyzed the data, contributed reagents/materials/analysis tools, authored or reviewed drafts of the paper, approved the final draft.

Victor Jesus Moreno-Alcaraz conceived and designed the experiments, performed the experiments, contributed reagents/materials/analysis tools, prepared figures and/or tables, authored or reviewed drafts of the paper, approved the final draft.

Maria Teresa Martinez-Romero performed the experiments, contributed reagents/materials/analysis tools, prepared figures and/or tables, authored or reviewed drafts of the paper, approved the final draft.

Alba Aparicio-Sarmiento performed the experiments, contributed reagents/materials/analysis tools, prepared figures and/or tables, approved the final draft.

Fernando Santonja-Medina conceived and designed the experiments, contributed reagents/materials/analysis tools, authored or reviewed drafts of the paper, approved the final draft.

The following information was supplied relating to ethical approvals (i.e., approving body and any reference numbers):

The study was approved by the Ethics and Research Committee of the University of Murcia (Spain) [ID: 1702/2017].

The following information was supplied regarding data availability:

The raw measurements are available in File S1.

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
