# Peer review of "Sagittal spinal morphotype assessment in 8 to 15 years old Inline Hockey players"

_PeerJ, doi:10.7717/peerj.8229_

## Round 0.1 · original submission · Minor Revisions

Dear Authors,

Your manuscript, "Sagittal spinal morphotype assessment in 8 to 15 years old Inline-Hockey players", has been reviewed by our Ad-hoc reviewers with expertise in the field who feel that your manuscript would be accepted for publication in Peerj before a minor revision. Their commentary is attached below for your information.

Thank you for the opportunity to review your manuscript at this time.

Sincerely,

Liang Gao, MD, PhD
Academic Editor
PeerJ — the Journal of Life and Environmental Sciences

Reviewer 1 ·

Basic reporting

The authors present a study on the assessment of sagittal spinal morphotype in young Inline-Hockey players. This study presented the assessment protocol of “Sagittal Integrative Morphotype” composing by the evaluation of sagittal spinal curvatures in a relaxed standing position, slump sitting position, and maximum flexion of the trunk. The protocol provided a diagnosis of sagittal spinal morphotype using the unilevel inclinometer. The author concluded that this method provided considerable reproducibility and validity with a good correlation with the radiographic measurement. Several issues need to be addressed before this work is suitable for publication.

Experimental design

1. Patients with radiographic evidence deformity may present a normal spinal morphotype, therefore, the limitations and shortcomings should be recognized and reflected upon.

2. Other reports assessing spinal morphology in young athletes to assist the preemptive care of early-stage spinal deformities need to be discussed.

3. How many observers are there in this study?

4. Quantitative statistics need to be performed to acknowledge the possible significant difference between groups.

Validity of the findings

1. The authors need to provide information/data regarding the interrater reliability of this protocol.

2. More clinical implications of the present study need to be highlighted.

Reviewer 2 ·

Basic reporting

The paper is well written, clear structured, and easy to follow. The introduction provides a good overview on previous works up to the state of the art. The hypotheses for the paper is clear defined. Raw data are provided with the manuscript. Only improvement I can see with the data is to provide it in English language.

Experimental design

The design of the experiment is adequate. Although it would be of interest to analyze the spinal shape changes with more recent techniques, such as wearable technologies that focus on the spinal shape and allow for long term measurement. The presented technique only takes a snapshot in lab conditions. Further, some more statistics about the distribution of the subjects into the age groups would be of interest. Similarly, it would be interesting how long the subjects are engaged in IH before the measurement was taken. This would help to see how the change in posture progresses. Regarding the statistical analysis it would be interesting to see a histogram of the data, as they are not normal distributed. Was any other defined distribution found?

Validity of the findings

Findings reported in this work are valid and sound.

Additional comments

This paper is in a good shape, I only have minor remarks that should be addressed, mentioned above.
In summary:
- convert raw data to English language.
- Add statistics about the distribution into the age groups.
- Add information about engagement time into IH, if available.
- Histogram or additional information about the data distribution.

---

## Round 0.2 · accepted · Accept

Dear authors,

I am glad to inform you that your article has been accepted to be published in the PeerJ.

Congratulations!

Best regards,

Liang Gao, MD, PhD
Academic Editor, The PeerJ

Reviewer 1 ·

Basic reporting

No comment.

Experimental design

No comment.

Validity of the findings

No comment.

Reviewer 2 ·

Basic reporting

The authors improved the manuscript in this revision, new tables have been added and the raw data have been translated to English.

Experimental design

Experimental design hasn't changed, but was fine in the first revision, already.

Validity of the findings

The authors have improved the article, and thus the findings are better supported, now.